# Translation of the Weight-Related Behaviours Questionnaire into a Short-Form Psychosocial Assessment Tool for the Detection of Women at Risk of Excessive Gestational Weight Gain

**DOI:** 10.3390/ijerph18189522

**Published:** 2021-09-09

**Authors:** Shanna Fealy, Lucy Leigh, Michael Hazelton, John Attia, Maralyn Foureur, Christopher Oldmeadow, Clare E. Collins, Roger Smith, Alexis J. Hure

**Affiliations:** 1School of Nursing, Paramedicine, and Healthcare Sciences, Faculty of Science and Health, Charles Sturt University, 7 Major Innes Road, Port Macquarie, NSW 2444, Australia; michael.hazelton@newcastle.edu.au; 2School of Medicine and Public Health, College of Health and Wellbeing, University of Newcastle, University Drive, Callaghan, NSW 2308, Australia; John.Attia@newcastle.edu.au (J.A.); roger.smith@newcastle.edu.au (R.S.); alexis.hure@newcastle.edu.au (A.J.H.); 3Hunter Medical Research Institute, Lot 1 Kookaburra Circuit, New Lambton Heights, NSW 2305, Australia; lucy.leigh@hmri.org.au (L.L.); christopher.oldmeadow@hmri.org.au (C.O.); clare.collins@newcastle.edu.au (C.E.C.); 4School of Nursing and Midwifery, College of Health and Wellbeing, University of Newcastle, University Drive, Callaghan, NSW 2308, Australia; maralyn.foureur@newcastle.edu.au; 5Hunter New England Health Nursing and Midwifery Research Centre, Newcastle, NSW 2300, Australia; 6School of Health Sciences, College of Health and Wellbeing, University of Newcastle, University Drive, Callaghan, NSW 2308, Australia; 7Department of Endocrinology, John Hunter Hospital, Newcastle, NSW 2305, Australia

**Keywords:** psychosocial, gestational weight gain, body image, weight gain attitudes, self-efficacy, pregnancy care, public health

## Abstract

The identification and measurement of psychosocial factors that are specific to pregnancy and relevant to gestational weight gain is a challenging task. Given the general lack of availability of pregnancy-specific psychosocial assessment instruments, the aim of this study was to develop a short-form psychosocial assessment tool for the detection of women at risk of excessive gestational weight gain with research and clinical practice applications. A staged scale reduction analysis of the weight-related behaviours questionnaire was conducted amongst a sample of 159 Australian pregnant women participating in the Women and Their Children’s Health (WATCH) pregnancy cohort study. Exploratory factor analysis, univariate logistic regression, and item response theory techniques were used to derive the minimum and most predictive questions for inclusion in the short-form assessment tool. Of the total 49 questionnaire items, 11 items, all 4 body image items, n = 4 attitudes towards weight gain, and n = 3 self-efficacy items, were retained as the strongest predictors of excessive gestational weight gain. These within-scale items were highly correlated, exhibiting high item information function value statistics, and were observed to have high probability (*p* < 0.05) for excessive gestational weight gain, in the univariate analysis. The short-form questionnaire may assist with the development of tailored health promotion interventions to support women psychologically and physiologically to optimise their pregnancy weight gain. Confirmatory factor analysis is now required.

## 1. Introduction

Globally, there has been a renewed focus on prioritising and promoting a healthy start to life, including appropriate weight gain in pregnancy [1]. In Australia, the revised Australian Department of Health Pregnancy Care Guidelines, released in 2018, have expanded their clinical assessment recommendations beyond overweight and obesity management (i.e., calculation of body mass index (BMI) and diet and physical activity advice), to highlighting the risks of excessive gestational weight gain (EGWG) at any pre-pregnancy BMI [2]. The revised guidelines now include consensus-based recommendations advising pregnant women to gain weight within the American Institute of Medicine (IOM) 2009 weight gain in pregnancy ranges [3], combined with routine antenatal weighing [2,3].

Dietary and physical activity habits during pregnancy and subsequent gestational weight gain (GWG) are important factors influencing both short- and long-term maternal and infant health status [4]. Excessive gestational weight gains (above the IOM weight gain in pregnancy reference ranges) have been associated with large for gestational infants, a greater odds for caesarean birth [5], pre-eclampsia or pregnancy-induced hypertension (PIH) [6,7], gestational diabetes mellitus (GDM) [7,8], and fatty acid composition of breastmilk [9]. Most concerning are the long-term and intergenerational chronic disease risks associated with EGWG highlighted by emerging research, including the Developmental Origins of Health and Disease (DOHaD) hypothesis and infant “gut” microbiome research [10].

The Developmental Origins of Health and Disease is a field of research that grew from pivotal work by Professor David Barker [11]. Barker hypothesised that early life exposure within the intrauterine environment may determine foetal physiological development and later life health outcomes [11]. Current understanding of the intrauterine environment, characterised by maternal over-nutrition and subsequent EGWG, is developing. However, research to date suggests that this may alter foetal physiology and the subsequent development of adult and childhood non-communicable diseases such as cardiovascular disease, diabetes and obesity [12].

The antenatal period provides a window of opportunity to promote positive heath behaviours, such as a nutritious diet that meets pregnancy nutrient reference values [4], and physical activity recommendations [13]; however, less is known about a woman’s psychosocial capacity for weight-related behaviour change during pregnancy [14]. 

The pregnancy experience, including weight gain, is highly variable, and influenced by a complex interplay between physiological, psychological, and sociological factors [15,16,17]. Psychosocial factors include body image, self-efficacy, locus of control, attitudes, beliefs, values, social support, depression and anxiety [18,19]. There is a growing body of research exploring the direct and indirect relationships between these psychosocial factors and heath behaviours [16,17], including their role in gestational weight gain (GWG) [18,19,20]. Although no cause and effect relationships have been established, cohort studies to date suggest that temporal relationships exist between psychosocial factors such as body image dissatisfaction (generally measured as satisfaction with weight or shape or attitudes towards external appearance) [21,22], depression, weight gain attitudes, social support and EGWG [18,19]. The majority of studies to date have employed observational designs such as cohort and cross-sectional [18,19]. Across studies, a variety of psychosocial constructs and measurement tools have been identified and evaluated for their relationships with EGWG [18,19]. A systematic review and narrative synthesis of 35 studies evaluating psychosocial and psychological antecedents of EGWG by Kapadia et al. [19] identified 26 different constructs as exposure variables. The number of identified constructs and variety of measurement tools were limitations of the review, with authors unable to pool studies using meta-analysis techniques [19].

There is a need to develop a consensus regarding psychosocial factors and scales of measurement that are predictive of EGWG [18,19]. A single tool that is quick to complete and relevant to clinical outcomes, similar to the Edinburgh Postnatal Depression Scale, may help with knowledge gains about weight gain in pregnancy [23]. The current focus for preventing EGWG is on lifestyle behaviours including diet and physical activity [24,25]. However, these targets for health behaviour change have yielded mixed results [24,25]. Therefore, there is a need to develop pregnancy-specific psychosocial measurement tools with broad research relevance and potential clinical application [16,26]. A single psychosocial assessment tool may offer new opportunities for health promotion and research during pregnancy.

### The Weight-Related Behaviours Questionnaire

Kendall et al. [27] developed and validated the Weight-Related Behaviours Questionnaire (WRB-Q), to assist with the identification of pregnancy-specific psychosocial factors affecting gestational weight gain and postpartum weight retention. The complete WRB-Q consists of 49 individual items measuring 6 psychosocial factors (subscales), using Likert scale responses [27]. The WRB-Q was developed without a global score or subscale scoring system.

The original WRB-Q was designed by combining existing psychosocial measurement tools from the available health behaviour literature [28,29,30] with qualitative study findings [31,32]. The WRB-Q was then tested and validated within the Bassett Mothers Health Cohort, a large (n = 622) prospective pregnant cohort study in the United States of America [27]. The WRB-Q subscales have been used to examine the relationships between psychosocial factors and outcomes such as GWG (excessive or inadequate) [33,34,35]. Across a series of research publications, Hinton et al. have explored the WRB-Q as a predictor of pregnancy and postpartum health behaviour including food intake and exercise frequency [36,37] and postpartum weight retention [38], primarily within one large American pregnancy cohort (Bassett mothers cohort, n = 622). Other studies have utilised selected WRB-Q subscales and have been cross-sectional in nature, conducted within Canadian (n = 330) [34] and Dutch samples of pregnant women (n = 258) [35]. It is unclear why the entire pregnancy-specific WRB-Q has not been used for research purposes more broadly. However, the use of selected subscales suggests that length of the complete questionnaire may not be practical for application, including in research settings. A systematic review of 38 studies, exploring the barriers and facilitators to shared decision making amongst health care professionals and their patients in clinical practice, found that time pressures were the most commonly identified barrier, featuring within 24 included studies [39]. Moreover, an integrated review investigating barriers to psychosocial assessment in hospital-based maternity care found that midwives and obstetricians reported a lack of time as a professional barrier and an organisational barrier, indicating a lack of willingness of institutions to allow for sufficient time for screening [40]. A study conducted by Ockenden et al. [41] additionally reported that psychosocial measurement tools developed prior to the release of the updated IOM 2009 weight gain in pregnancy guidelines such as the WRB-Q could be perceived as outdated, limiting its use within the published literature. Therefore, the primary aim of the present study was to determine which of the WRB-Q items are most suited for inclusion into a short-form pregnancy-specific psychosocial assessment tool.

## 2. Materials and Methods

### 2.1. Study Design

This was a scale reduction analysis using exploratory factor analysis (EFA), univariate logistic regression and item response theory (IRT) techniques. Weight gain and WRB-Q data were collected from participants within the Women And Their Children’s Health (WATCH) pregnancy cohort study [42]. A pragmatic staged-design approach was undertaken, as displayed in Figure 1. The scale reduction process was guided by the pragmatic research paradigm applied widely within social science research, whereby practical problem-solving techniques are employed [43]. All analyses were performed using STATA 14.0 (StataCorp LLC, College Station, TX, USA) and SAS V9.4 (SAS Institute Inc., NC, USA), by a statistician who was blinded to the original data collection.

### 2.2. Population Sample and Data Collection

The sample for the analysis was drawn from the WATCH study. The WATCH study was a small (n = 180 women and n = 182 children) Australian prospective longitudinal study, where women were recruited to the study if they were <18 weeks gestation, with follow-up occurring until 4 years post birth [42]. Women were recruited to the cohort between June 2006 and December 2007. Pregnancy and weight data were collected during antenatal care visits by researchers at approximately 19, 24, 30 and 36 weeks gestation. The 49 item WRB-Q was self-administered to participants at the first study visit where participants were approximately 19 weeks gestation. The questionnaire response rate was 88%, completed by n = 159 WATCH participants. Maternal pre-pregnancy weight (kilograms) was self-reported at the first study visit only. All subsequent weight measurements were conducted by researchers, who held Level I anthropometry qualifications. Total GWG was calculated by subtracting the last recorded pregnancy weight at approximately 36 weeks gestation, from the self-reported pre-pregnancy weight measurement as per the detailed study paper [42]. The research protocol for the WATCH study was approved by the Hunter New England Health Human Research Ethics Committee (approval number 06/05/24/5.06).

### 2.3. WRB-Q Items and Scales of Measurement

The 6 psychosocial subscales are:Weight locus of control (WLOC)—4 questionnaire items (5-point Likert scale ranging from “strongly agree to strongly disagree” with a neutral option where 3 indicates “neither agree or disagree”), indicating whether a woman feels she has control over her body weight (internal WLOC) or if body weight is something a woman feels she has little control over (external WLOC);Self-efficacy—8 questionnaire items (5-point Likert scale, “very sure to very unsure”, where 3 indicates “neither sure or unsure”), indicating levels of confidence for diet, exercise and postpartum weight loss behaviour change;Attitudes towards weight gain—13 questionnaire items (5-point Likert scale, “strongly agree to strongly disagree”, where 3 is “neither agree or disagree”), indicating personal attitudes towards gaining weight during pregnancy or weight gain avoidance;Body image—4 questionnaire items (2 items—4-point scale ranging from “very satisfied to very dissatisfied” with no neutral option and 2 items—reported on a scale of ”too heavy, about right and too light), indicating personal satisfaction with body weight and shape and perception of body weight and shape;Feelings about the motherhood role—7 questionnaire items (5-point Likert scale, “strongly agree to strongly disagree”, where 3 is “neither agree or disagree”), indicating positive and negative perceptions of motherhood;Career orientation—13 questionnaire items (4-point Likert scale “strongly agree to strongly disagree”, no neutral option), indicating preference towards career or family [27].

### 2.4. Scale Reduction Analysis

#### 2.4.1. Stage 1

Exploratory factor analysis (EFA) with principal axis factoring and varimax rotation was performed for all WRB-Q items listed under the 6 psychosocial subscales, to examine their overall performance (i.e., construct validity and internal consistency) within the WATCH cohort. The results of this analysis have been reported elsewhere [44]. Briefly, the EFA conducted amongst the WATCH sample indicated that the weight locus of control, self-efficacy and body image scales demonstrated consistent construct validity, retaining the same item factor structure to the original analysis conducted by Kendall, Olson and Frongillo [27]. All 6 psychosocial subscales demonstrated acceptable internal consistency (Cronbach’s alphas α > 0.70) [45], when tested amongst the WATCH cohort with the exception of the weight locus of contro*l* scale (Cronbach’s α 0.49) [44]. In the current analysis, EFA was a necessary step in the scale reduction process, accounting for the assumption of unidimensionality (i.e., checking that the data are appropriate for the model), for the application of further analysis techniques such as IRT [46]. During the EFA, strength of scale item correlations was examined using Spearman’s rho (*r_s_*) coefficient and *p* values (<0.05) [44], indicating the presence of probable redundant items. To detect these redundant items, further deleted items modelling analysis was performed on each of the 6 psychosocial scales. Where the deletion of scale items improved the internal consistency (Cronbach’s alphas α) of each individual scale, these were considered as redundant items, as displayed in Figure 2 [45]. These results were the precursor for reducing the WRB-Q into a short form [44].

#### 2.4.2. Stage 2

Univariate logistic regression was conducted to examine the relationship between each of the individual WRB-Q items and EGWG (measured at approximately 36 weeks gestation). The strength of associations was assessed via the magnitude of the odds ratios (OR) and statistical significance (*p*-values < 0.05). Four individual questionnaire items (items 7, 8, 9, 10 and 36) violated the assumption of linearity (between the predictor and outcome) and were examined and reported using categorical analysis techniques. Due to multiple hypothesis testing, Hochberg False Discovery Rate (FDR) procedures were applied to account for type 1 error [47].

#### 2.4.3. Stage 3

Item response theory (IRT) is a collection of techniques that is increasingly being applied to the development of questionnaire instruments or shortening of existing instruments as part of scale reduction [46]. Item response theory can evaluate the relationship between a person’s response to a particular questionnaire item and the level of construct being measured, essentially assessing the quality of scale items [46]. In this analysis, IRT was used for the purposes of scale reduction to further assess the quality of the WRB-Q items and test the strength of associations between the WATCH cohort questionnaire responses (measured by Likert scales) and the respective WRB-Q psychosocial subscales (latent trait variables). For dichotomous data, IRT models produce a trace line called the item characteristic curve (ICC) defined by the location (*a* parameter) and the slope (*b* parameter) and provide a visual representation of item performance and a corresponding value statistic [46]. For polytomous data such as the WRB-Q where questionnaire items are ordered categorical Likert scale items, a graded response model (GRM) of IRT was deemed the most appropriate [46]. The GRM utilises cumulative logistic regression to relate each questionnaire item to its respective psychosocial scale, essentially modelling the probability of a lower item response vs. a higher item response (e.g., scoring a 1 vs. a 2, 3, 4, or 5 or scoring 1 or 2 vs. a 3, 4 or 5, or scoring a 1, 2 or 3 vs. a 4 or 5, etc). A participant’s response to each item depends on both their “ability” (i.e., their level of construct—specific to each person), as well as the difficulty (*b* parameter) and discrimination (*a* parameter) of the item [46]. The discrimination parameter is an indication of how well an item is at differentiating between individuals with different levels of the latent trait [46]. All questionnaire items supply differing amounts of information towards their respective subscale; the amount of information an item contributes is dependent on the item’s discrimination and difficulty. There is no predefined cut-off point for the discrimination; rather, the values should be interpreted relative to other items on the same subscale. For the current analysis, graphical item information functions (IIF) were generated for each of the 6 psychosocial subscales. Essentially, items exhibiting peaked IIF curves will have a higher value statistic compared to other scale items, indicating that the item does a better job at supplying information than other items within the scale [46]. 

The standard GRM (in which a unique discrimination parameter is estimated for each item) did not converge for the body image scale. To address this, an alternative constrained GRM was performed, in which the discrimination parameters of items 26 and 27 were constrained to be equal, and items 28 and 29 were constrained to be equal. This was performed using SAS.

#### 2.4.4. Stage 4

In this stage, all analyses were considered in unison, with only items, that consistently performed well across all analyses (EFA, logistic regressions, and IRT) considered as candidates for inclusion within the gestational weight gain psychosocial assessment tool.

## 3. Results

Demographic and weight gain characteristics of the WATCH sample have been published previously [21,44]. Briefly, participants’ mean age was 28.9 years (SD 5.64), 71% had an education level at or above high school completion, 61% were married, and 55% were multiparous. Weight gain data were available for 147 participants. Fifty-one percent of participants (n = 75) recorded a pre-pregnancy BMI in the normal weight category (≥18.5–24.9 kg/m^2^); 5% (n = 8) recorded a pre-pregnancy BMI in the underweight category (<18.5 kg/m^2)^; 23% (n = 34) recorded a BMI in the overweight category (≥25–29.9 kg/m^2^), with 20% of participants (n = 30) recording a pre-pregnancy BMI in the obese category (≥30 kg/m^2^). Of these, 60 participants had gained weight above the IOM GWG reference values by the time they reached 36 weeks gestation (BMI < 18.5 kg/m^2^, n = 5; BMI ≥ 18.5–24.9 kg/m^2^, n = 24; BMI ≥ 25–29.9 kg/m^2^, n = 20; BMI ≥ 30 kg/m^2^, n = 11) [21].

### 3.1. Stage 1

Results of the deleted items modelling (Cronbach’s alpha (α)) conducted as part of the EFA are presented in Figure 2. Deleted items modelling revealed that the internal consistency of the psychosocial scales could be improved with the deletion of selected items. Where the deletion of items strengthened the internal consistency of a psychosocial subscale, these were labelled as “DROP” items, with all other well-performing items labelled as “KEEP” items.

### 3.2. Stage 2

Univariate logistic regression results identified 13 individual items across four psychosocial subscales as predictors of EGWG (*p* < 0.05), as displayed in Figure 2. These included all Body image scale items (items 26–29, *p* <0.01); four Self-efficacy items (Items 8, 9, 10, and 12, *p* < 0.05); four items from the Attitudes towards weight gain scale (items 13–16, *p* <0.05); and one item from the Career orientation scale (Item 44, *p* <0.05). Following false discovery rate adjustments, none of the questionnaire items demonstrated a statistically significant relationship with EGWG. As the determination of this relationship was not the primary objective of the study, the unadjusted univariate results were used to guide item selection. Only items exhibiting high probability relationships with EGWG (*p* < 0.05) were considered for inclusion in the assessment tool.

### 3.3. Stage 3

The graphical item information function (IIF) results are presented in Appendix A. Item information function value statistics, for all WRB-Q items ranged between −0.11 and 8.80, as displayed in Figure 2.

Descriptions of the IIFs for each psychosocial scale are as follows. For the weight locus of control scale, item 2 contributed the greatest amount of information, having the highest estimated discrimination value statistic (4.12). Item 1 provided some information (1.72), with items 3 and 4 contributing very little information (exhibiting flat curves) with low value statistics. Of the self-efficacy scale, item 10 contributed the most information, exhibiting the highest value statistic (2.77), followed by item 9 (2.49) and item 8 (2.38). For the attitudes towards weight gain scale, item 13 contributed the most information (3.44), followed by items 15 (3.17), 14 (3.08) and 16 (2.84). All body image scale items exhibited high discrimination values, with the highest value observed for items 26 and 27 (8.80), followed by items 28 and 29 (3.31). As the discrimination was constrained to be equal, the IIF plots for items 26 and 27 and items 28 and 29 are identical as per Appendix A. For the feelings towards the motherhood role scale, item 33 (2.46) exhibited the highest discrimination value followed by item 32 (2.34) and item 31 (1.43). For the career orientation scale, item 38 exhibited the highest discrimination value (1.87) followed by item 40 (1.60) and item 39 (1.56).

### 3.4. Stage 4

As per Figure 2, when the EFA, univariate analysis and IRT were taken together, a total of 11 items across three psychosocial scales (self-efficacy, attitudes towards weight gain, body image) were determined as candidates for inclusion (i.e., consistently performing well across all stages of analysis) in a short-form assessment tool. As displayed in Table 1, three self-efficacy items (items 8, 9, 10), exhibited high probability for predicting EGWG (*p* < 0.05). These same items were all highly correlated with each other (i.e., loading on the same factor) and contributed the most information to the scale (high IIF value statistics). Four items from the Attitudes towards weight gain scale (items 13, 14, 15, 16) exhibited high probability for predicting EGWG (*p* < 0.05). These same items were again highly correlated with each other and had high discrimination value statistics. All body image scale items were predictive of EGWG (*p* < 0.01), highly correlated with each other and exhibited high IIF value statistics.

## 4. Discussion

The current study evaluated results of a scale reduction analysis of the WRB-Q originally developed by Kendall, Olson and Frongillo [27] for use in pregnancy. Our analysis has furthered this body of work, identifying 11 questionnaire items performing consistently well across all analyses as candidates for combination into a short-form assessment tool to predict EGWG. Shortening the WRB-Q from 49 items across six psychosocial subscales to 11 items across three subscales with high predictive value for EGWG may increase the questionnaire’s utility for both research and clinical practice application. This new analysis was conducted within a contemporary Australian pregnancy cohort [42], using the IOM 2009 weight gain ranges, ensuring that only those psychosocial factors relevant to current public health guidance have been identified.

Of the 11 questionnaire items selected from the full WRB-Q, 8 items were specifically related to weight shame and/or body image dissatisfaction. The unadjusted univariate analysis results indicated that higher body image scores (indicating greater satisfaction with body image) were associated with a decreased odds of experiencing EGWG. The questions related to weight shame (items 13–16), indicating attitudes towards weight gain avoidance, were also associated with greater odds of EGWG. These results suggest that some women might benefit from tailored care approaches that seek to reduce weight shame and embarrassment and improve body image satisfaction during pregnancy. The remaining questionnaire items with high probability for predicting EGWG were derived from the self-efficacy scale (items 8–10). These items specifically addressed perceived confidence towards diet and food intake, with higher perceived self-efficacy scores towards eating food that is “good for you” and avoiding foods that are not good for you, associated with a lower odds of EGWG. These results indicate that some women may need and benefit from more support in eating a more balanced diet, particularly those with a busy family/work life, with further research needed to evaluate the outcomes of a psychosocial risk-based approach to diet required. These findings are of particular interest given that current weight management guidance in Australia aims to prevent EGWG with a physiological focus, through healthy eating and physical activity advice and weight monitoring [2], without much emphasis on the cause of EGWG or the longer term and intergenerational effects of EGWG on maternal and infant health.

A recent review and discussion of maternal body image dissatisfaction in childbearing and early childhood reported that body image is an important but often overlooked psychosocial factor that mediates (barrier/enabler) weight gain in pregnancy [16]. A recent Australian cross-sectional study by Fealy et al. [21] found body image to be predictive of EGWG. In this study, for every one unit increase in body image score, a 33% decreased odds of EGWG was observed [21]. The rapid physiological changes that occur for body shape, weight and size during pregnancy and an evident bi-directional relationship between body image and depression (i.e., body image dissatisfaction increases the risks of depression and depression increases the risks of body image dissatisfaction) indicate the need to broaden current psychosocial screening to include satisfaction with body image [16,26,48]. Moreover, Dryer et al. [26] assert that given the rapid physiological changes to body shape, weight and size that occur during pregnancy, health professionals need to evaluate body image to increase their awareness and responsiveness to women’s psychosocial needs, so as to not exacerbate or contribute to the development of pregnancy-specific anxiety, depression or disordered eating, particularly given that weight shame is still prevalent amongst health professionals [26,49].

Pregnant women have described their experiences of GWG with health care professionals as stressful, confusing and judgmental [50,51]. This, coupled with a lack of clinical guidance, appropriately qualified health professionals and focus on diet, exercise, and weight gain, may contribute to negative health behaviours such as disordered eating, low self-esteem and social exclusion [13,49]. A systematic review and qualitative synthesis by Vanstone et al. [50] reported that when women received nutritional and physical activity advice from health care providers, the advice rarely considered their individual circumstances. Women consistently reported significant social and economic disadvantages as barriers to healthy eating/and physical activity, with authors arguing that it is unethical to directly target the physiological aspects of weight gain alone [50]. By evaluating the psychosocial factors from this short-form WRB-Q, such as body image and attitudes towards weight gain, early in pregnancy, researchers and health care professionals may better understand the motivation, readiness and capacity for health behaviour change [52]. Health promotion approaches, delivered by appropriately qualified health professionals who are considerate of a woman’s psychosocial factors, that aim to reduce weight shame, improve body image satisfaction and improve eating habits, could increase adherence to gestational weight gain targets, improve health professional engagement and increase women’s satisfaction with this aspect of maternity care.

Confirmatory factor analysis amongst a large independent pregnancy cohort is now needed to assess the construct validity and internal constancy of the short-form assessment tool. It is hoped that by reducing the WRB-Q into a short form, specifically for the detection of women at risk of EGWG, research in this area may increase and allow for the eventual pooling of results by meta-analysis techniques to confirm these relationships. The eventual translation of the assessment tool into real world maternity care practice could genuinely support women to achieve healthy weight gain during pregnancy.

### 4.1. Strengths

This paper proposes a short-form WRB-Q tool to assess psychosocial factors that may be useful in predicting EGWG. Further testing is now needed to confirm the performance (reliability and validly) of the short form within larger and more diverse cohorts of pregnant women. The short-form WRB-Q may go some way to reduce the burden of time for participants and researchers and may be more practical for use in both clinical research and practice settings than the original WRB-Q.

### 4.2. Limitations

Due to the small sample size, multi-dimensional IRT, which would also take into account the multi-factor structure within each subscale, was not performed. IRT generally requires large sample sizes (n = 100 s to 1000 s) for adequate analysis. However, Edelen et al. [46] argue that variables can be adequately tested within samples of between 200 and 500 subjects, and that questionnaire properties can be assessed with sample sizes as small as <100 subjects. Sahin et al. [53], in a study investigating the effects of test length in terms of questionnaire items and sample size on IRT parameters, found that a sample of as low as n = 150 participants could be used with tests of 10, 20, or 30 items to accurately estimate the tests statistic. The authors also cautioned that the use of small sample sizes of approximately n = 150 may still be biased and that caution is needed with interpretation [53]. In the current analysis, the current subscale scores are based on a summation of items. A potential improvement on this method would be to use IRT to create continuous subscale scores, which are potentially more easily utilised in subsequent analysis. However, such analyses usually require upwards of 500 subjects to ensure accurate parameter estimates [46] and so was not possible in the current study. Given these limitations, we have attempted to reduce the potential bias due to the smaller sample by using three sets of results (EFA, logistic regressions, and IRT).

No universal or subscale summary scores predicting risk of EGWG were calculated for the newly developed 11 item short form. This was considered outside the scope of the current analysis and, as such, is a limitation of the study. The data informing this analysis were based on the WRB-Q administered at one time point during mid-pregnancy and, as such, we are unable to determine the scales’ performance or predictive value earlier or later in pregnancy. Pre-pregnancy BMI was calculated by using a combination of self-reported weight and objective (clinician measured) height reference values. We acknowledge that the use of self-reported weight is a potential limitation of the current study. Additionally, the study was conducted amongst one small Australian cohort from one hospital and, as such, the results may not be generalizable to other more culturally diverse cohorts of Australian women. Australian women may not hold the same attitudes to weight gain as other nationalities and, hence, this requires further testing amongst nationally and internationally representative cohorts of pregnant women.

## 5. Conclusions

These analyses have produced a short-form psychosocial assessment tool for the screening and detection of women at risk of experiencing EGWG. Collectively assessing these psychosocial factors using the newly developed assessment tool may go some way to assist with the design and development of tailored health promotion interventions that support women psychologically and physiologically to optimise their pregnancy weight gain. Further testing of the short-form questionnaire by confirmatory factor analysis is now needed to progress research in this area.

## Figures and Tables

**Figure 1 ijerph-18-09522-f001:**
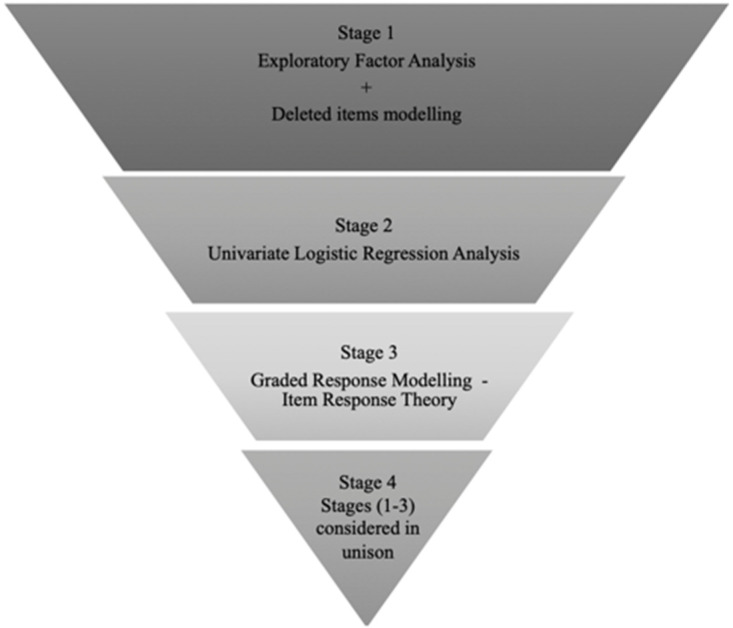
Scale reduction study design for a short form questionnaire.

**Figure 2 ijerph-18-09522-f002:**
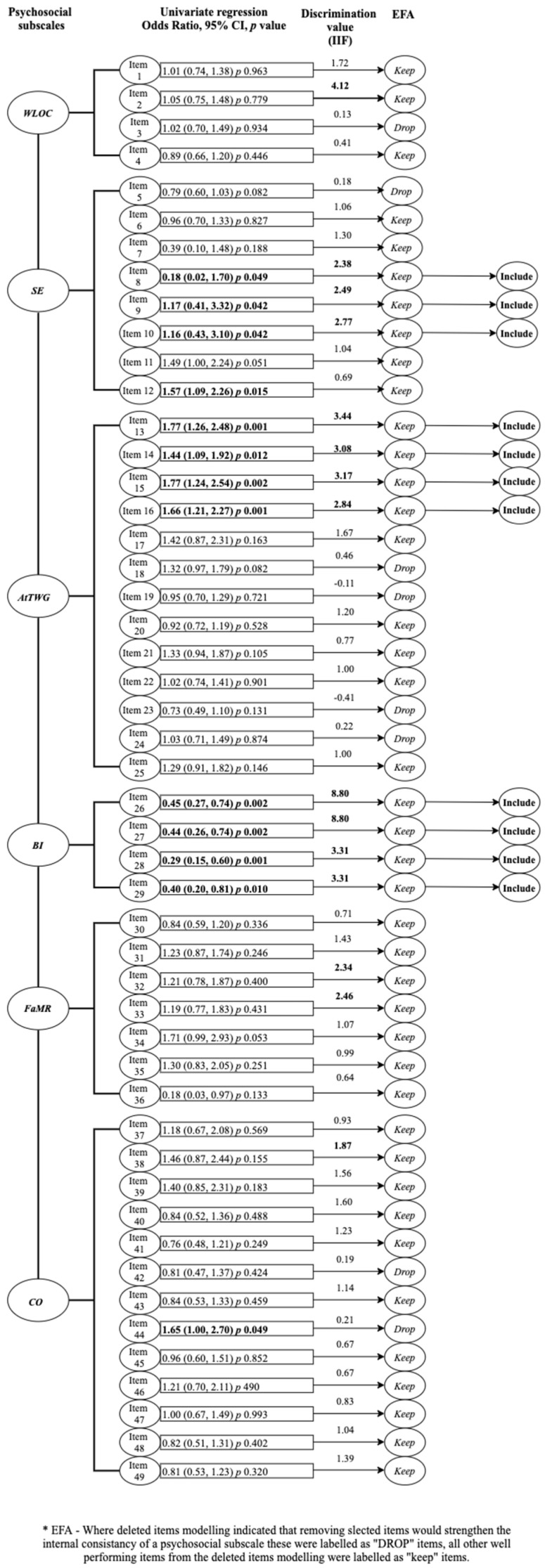
Full model of results from the scale reduction analysis.

**Table 1 ijerph-18-09522-t001:** Gestational Weight Gain Psychosocial Risk Assessment Tool.

Self-EfficacyHow Sure Are You That You Can?	Very Sure	Sure	Neither Sure nor Unsure	Unsure	Very Unsure
Eat balanced meals	1	2	3	4	5
Eat foods that are good for you & avoid foods that are not.	1	2	3	4	5
Eat foods that are good for you even when family or social life takes a lot of your time…	1	2	3	4	5
**Attitudes towards Weight Gain** **Circle the Response That Best Represents How You Feel:**	**Strongly Agree**	**Agree**	**Neither Agree nor Disagree**	**Disagree**	**Strongly Disagree**
The weight I gain during my pregnancy makes me feel ugly	1	2	3	4	5
I worry that I may get fat during this pregnancy.	1	2	3	4	5
I am embarrassed at how big I have gotton during this pregnancy.	1	2	3	4	5
I’m embarrassed whenever the nurse weighs me.	1	2	3	4	5
Body Image	
**Circle the Response That Best Represents How You Feel:**	**Very Satisfied**	**Satisfied**	**Dissatisfied**	**Very Dissatisfied**	
How satisfied are you with your current shape?	0	1	2	3	
How satisfied are you with your current weight?	0	1	2	3	
	**Too Heavy**	**About Right**	**Too Light**		
Do you consider your current weight to be…	0	1	2		
Do you consider your current body shape to be…	0	1	2

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
