# Peer review of "Translation of the Weight-Related Behaviours Questionnaire into a Short-Form Psychosocial Assessment Tool for the Detection of Women at Risk of Excessive Gestational Weight Gain"

_ijerph, 2021, doi:10.3390/ijerph18189522_

Round 1

Reviewer 1 Report

In this paper, the authors proposed a short-form WRB-Q tool to assess psychosocial factors useful in predicting EGWG. They proposed that it will reduce the burden of time and will be more practical for use in both clinical research and practice settings than the original WRB-Q.

The authors mention in the abstract “A staged scale reduction analysis of the 24 weight related behaviours questionnaire was conducted amongst a sample of 159 Australian pregnant women participating in the Women And Their Children’s Health (WATCH) pregnancy cohort study. Exploratory factor analysis, univariate logistic regression, and item response theory techniques were used to derive the minimum and most predictive questions for inclusion in the short form assessment tool. 11 questionnaire items from the body image, attitudes towards weight gain and self-efficacy psychosocial scales were the strongest predictors of excessive gestational weight gain, deemed suitable for combination into the short form. The short-form questionnaire may assist with development of tailored health promotion interventions that support women psychologically and physiologically to optimise their pregnancy weight gain. Confirmatory factor analysis is now required.” However, it is found that the results are not discussed in the abstract.

In line 244: The sentence “Seventy-five participants recorded a pre-pregnancy BMI of” is incomplete. If it is “Seventy-five participants recorded a pre-pregnancy BMI of between ≥18.5 – 24.9 kg/m2….”, the sentence needs proper reconstruction.

The article has described the methods in detail which is impressive. However, a portion of the results was already published in earlier articles (as referenced by the authors in the result section as “Demographic and weight gain characteristics of the WATCH sample have been previously published [7,34].”). In addition, the present article does not convey the results clearly. There is no table or simplified figure that shows the findings of the study clearly. The manuscript, although written elaborately, failed to convey the results concisely and in a simplified manner. Overall, the article is difficult to follow and the results are difficult to interpret. A portion of the results is also published earlier in other articles referenced and acknowledged by the authors themselves. The authors might consider presenting the findings of the present study in a simplified table and figure as well, apart from the figure 2 already provided.

The conclusion from the study is neither robust nor confirmatory as acknowledged by the authors by the use of phrases like “may be used to screen..” and “may go some way to assist…”

Reviewer 2 Report

Manuscript ID: ijerph-1322328

Title: Translation of the weight related behaviors questionnaire into a short-form psychological assessment tool for the detection of women at risk of excessive weight gain

Brief Summary

The aim of this study was to develop a short psychosocial assessment tool to detect women at risk for excessive gestational weight gain.  Researchers conducted a staged reduction analysis of an existing questionnaire, the Weight Related Behaviors Questionnaire, in a sample of 159 Australian women in the WATCH pregnancy cohort study.  Exploratory factor analysis, univariate logistic regression, and item response theory techniques were used to distill down to the minimum number of items deemed most predictive.  This resulted in 11 items from the body image, attitudes towards weight gain, and self-efficacy subscales of the original questionnaire being included in the short form of the questionnaire.

General comments

  • The development of a shorter version of the Weight Related Behaviors Questionnaire represents a useful contribution to efforts to reduce excessive gestational weight gain. However, there are some limitations and issues present in the manuscript which require further attention.
  • While generally well-written, there are mistakes in the in-text citations and reference list that need correcting.

Specific comments

Title: Weight Related Behaviors Questionnaire should be capitalized as it is the name of a specific questionnaire, thus a proper noun.  Also, the title should specify “excessive gestational weight gain” to reflect that the study was about weight gain during pregnancy.

Abstract

Line 29: Spell out numbers at the beginning of a sentence.

  1. Introduction

Lines 46-48:  The authors reference the Institute of Medicine Guidelines from 2009 but do not provide a reference for the Guidelines themselves.  Please add to the in-text citations and to the reference list.

Line 48:  Please correct the reference for in-text citation #4 in the reference list. The title is incorrect (should be Weight Gain During Pregnancy, not Nutrition During Pregnancy…)and the reference is incomplete, missing book title, publisher, publishing location, and specific page numbers referenced (see IJERPH guidelines for book references).

Line 50: Please use the correct reference for in-text citation #6.  The reference listed is for a commentary on an article, not the actual peer-reviewed article itself.

Lines 51-61:  The information presented here suggests pregnant Australian women are not eating enough rather than too much, so it is unclear why this is included considering the point of the study is to prevent excessive gestational weight gain.  This contradicts the authors’ claim that excessive gestational weight gain is a problem in Australia.

Line 53-57:  This sentence is missing a citation.  Here and elsewhere in the paper, put the citation number, not the publication year, after naming a specific researcher within the text as follows:

“A recent study by Slater et al. [5]…”

Line 58:  “servers” should be “servings”.

Lines 59-61 and lines 62-63:  Please replace reference #7 with references that match the statements.  The cited paper by Fealy et al. reported on demographic and social-cognitive factors related to gestational weight gain, not short and long term implications of poor maternal nutrition (lines 59-61). Also, the paper does not discuss physiological or psychological factors related to weight gain during pregnancy (Line 73).

Lines 69-70:  Please define “body image dissatisfaction” and provide a citation.  Dissatisfaction with what specifically?  Body shape, size, too fat, too thin?

Line 75-76:  Instead of listing all seven authors, write “Kapadia et al. [9]…”.

Line 76:  Please check the referenced article for the correct number of constructs reported.

Line 90: “et al.” does not need to be italicized. 

Line 84-85:  Actually, the Cochrane review (ref #15) concluded that the evidence for the benefit of diet and/or exercise in reducing gestational weight gain was of “high quality”. Ref #16 looked at interventions with a nutritional component  aimed at managing gestational weight gain and postpartum weight retention and found mixed results, which is not the same as “modest results at best”.  Thus, “questioning” the role of targeting diet and physical activity to prevent gestational weight gain is unwarranted.  Please revise or omit this statement.

Lines 85-87:  Refs # 10 and 17 regard body dissatisfaction, which is already included in existing psychosocial measurement tools.  How do these references support the statement that “there is a need to develop pregnancy specific psychosocial measurement tools…”?

Lines 101-105.  Please rewrite or separate out Ref #29 from this sentence that begins “Hinton and Olson (2001)…”.  Ref #29 is from 2016 and does not include Hinton as a co-author.

Lines 108-111:  The authors suggest a valid argument for the need for a shorter-form questionnaire.  However, the study cited has limited value as a reference; it only reported on Australian midwives in private hospitals who perceived time  constraints as a barrier to psychosocial assessment and depression screening.  Suggest finding and citing other research regarding barrier (time constraint) of lengthy questionnaires as perceived by patients or health-care providers in wider settings.  Also, please rewrite this sentence as it is very long and clunky as written.

  1. Materials and Methods

Line 183: Please spell out abbreviation at first use in the text, “item response theory (IRT)”

Line 194:  “…Odd Ratios…” should be spelled with lower case letters.

Lines 205-209:  Please explain why IRT was considered appropriate for this analysis with your very small sample size (n=159), considering most studies using IRT have sample sizes in the 1000s.  Include references to support your argument.

  1. Results

Line 185:  Ref# 36 does not seem related to the statement it follows.  Please use the correct reference here and use Ref #36 where Cronbach’s alpha is first introduced earlier in the manuscript (Line 179).

Line 244: “Weight gain data were…”(not was). Data are plural.

  1. Discussion

Lines 336-338: As this statement is not relevant to the topic addressed in this paper, please omit.

Line 341: Here and elsewhere in the paper, put the citation number, not the publication year, after naming a specific researcher.

Line 343-344:  Summarize results here; OR, CI and p-values are not reported in the Discussion section.

Lines 345-348: Body image should be spelled out, not abbreviated.

Lines 348-349:  For more than two authors, use [author name] et al.

Lines 356-359: “…may contribute to negative health behaviors such as disordered eating, low self-esteem, and social exclusion.” Please provide citations to back up this statement.

Lines 359-360: Same comment as for Lines 348-349.

Line 378:  Instead of using a semicolon here, just start a new sentence.

Line 361: Suggest replacing the word “discussed” in this sentence. Systematic reviews and qualitative syntheses (subject) don’t “discuss” (verb).

Line 362: To what does ‘it” refer? 

Line 388: Suggest reviewing limitations.  Please describe concerns related to the small sample size and the concerns that the sample has very limited generalizability.  Given that the women in the study were patients from one hospital in Australia, the results may or may not be generalizable to other women in Australia.  Results are likely not generalizable to women in other countries, cultures, etc. Example: Do Australian women have the same attitudes toward EGWG as African American women?  Cross-national studies would need to be performed and can be suggested as an area for further research.

References (as numbered)– Recommend double checking the references.  Some are missing information.

  1. Year accessed?
  2. Reference incomplete as written
  3. Incorrect title
  4. Some unnecessary information in journal name
  5. Page numbers should be 5-18

Author Response

Please see that attachment 

Reviewer 3 Report

The manuscript is very interesting and well written. The structure and methodology is adequate. The authors study an important aspect during pregnancy. Especially due to the increased incidence of overweight and obesity during pregnancy. However, I have the following comments (minor comments).

I. Minor comments:
1. I suggest in the introduction and discussion, to present antecedents regarding how diet influences body weight during pregnancy, and the nutrient composition of breast milk. Especially fatty acids.
Suggested reference:
The Impact of Maternal Diet during Pregnancy and Lactation on the Fatty Acid Composition of Erythrocytes and Breast Milk of Chilean Women. Nutrients. 2018; 10: 839.

2. It would be interesting, in the introduction, to write a brief paragraph regarding the adverse metabolic consequences for fetal development, associated with excessive weight gain during pregnancy.

3. Improve the writing of the objective of the study.
4. Increase the font size in figure 2.
5. Is it possible to increase the size and resolution of figure 2 ?, it was difficult for me to read.
6. Improve the resolution of table 1.

Round 2

Reviewer 1 Report

All my concerns have been addressed. I have no further comments. Thank you.